# As Easy as 1, 2, 3? How to Determine CBCT Frequency in Adjuvant Breast Radiotherapy

**DOI:** 10.3390/cancers14174164

**Published:** 2022-08-27

**Authors:** Yannis Junker, Mathias Düsberg, Rebecca Asadpour, Sophie Klusen, Stefan Münch, Denise Bernhardt, Stephanie Elisabeth Combs, Kai Joachim Borm

**Affiliations:** 1Department of Radiation Oncology, Klinikum Rechts der Isar, Technical University Munich, 81675 Munich, Germany; 2German Cancer Consortium (DKTK), Partner Site Munich, and German Cancer Research Center (DKFZ), 69120 Heidelberg, Germany; 3Institute of Radiation Medicine (IRM), Helmholtz Zentrum München, German Research Center for Environmental Health GmbH, 85764 Neuherberg, Germany

**Keywords:** adjuvant breast radiotherapy, image-guided radiation therapy, cone-beam computed tomography

## Abstract

**Simple Summary:**

Despite its crucial impact on treatment accuracy and therefore patient safety, image-guided radiation therapy (IGRT) during adjuvant breast radiotherapy widely lacks standardized recommendations and guidelines. A wide variation of different IGRT methods and schedules is used in radiotherapy centers over the world, yet reliable evidence remains sparse for most of these approaches. In this study, we took a closer look at a common approach, by which IGRT frequency is altered based on the magnitude of setup errors in the beginning of the radiotherapy treatment.

**Abstract:**

The current study aims to assess the suitability of setup errors during the first three treatment fractions to determine cone-beam computed tomography (CBCT) frequency in adjuvant breast radiotherapy. For this, 45 breast cancer patients receiving non-hypofractionated radiotherapy after lumpectomy, including a simultaneous integrated boost (SIB) to the tumor bed and daily CBCT imaging, were retrospectively selected. In a first step, mean and maximum setup errors on treatment days 1–3 were correlated with the mean setup errors during subsequent treatment days. In a second step, dose distribution was estimated using a dose accumulation workflow based on deformable image registration, and setup errors on treatment days 1–3 were correlated with dose deviations in the clinical target volumes (CTV) and organs at risk (OAR). No significant correlation was found between mean and maximum setup errors on treatment days 1–3 and mean setup errors during subsequent treatment days. In addition, mean and maximum setup errors on treatment days 1–3 correlated poorly with dose coverage of the CTVs and dose to the OARs. Thus, CBCT frequency in adjuvant breast radiotherapy should not be determined solely based on the magnitude of setup errors during the first three treatment fractions.

## 1. Introduction

The use of increasingly complex target volumes and highly conformal irradiation techniques in breast cancer radiotherapy requires precise position verification and adjustment through image-guided radiation therapy (IGRT), now more than ever. In recent years, body-surface scanners emerged as an alternative imaging method to cone-beam computed tomography (CBCT) and planar imaging. Yet, recent studies confirmed CBCT as the gold standard for IGRT as it reduces setup errors and allows for a superior spatial match with the planning CT [1,2]. The benefits of CBCT over other imaging techniques become apparent with increasing complexity of treated target volumes (e.g., regional nodal irradiation [RNI]) and irradiation techniques (e.g., SIB) [3].

A substantial drawback of CBCT is the increased imaging dose administered to the patient, especially with daily imaging schedules [4]. Several methods of CBCT dose reduction have been studied, and a shortening of scan exposure time and beam rotation range have led to a significant decrease in healthy tissue doses [5]. However, the most effective method to decrease additional dose exposure due to CBCT imaging is a reduction in CBCT frequency. A previous analysis by our study group indicates that even in case of complex target volumes including SIB irradiation, CBCT frequency should be determined on a case-by-case basis, instead of using daily imaging as standard procedure [6]. However, the question of how to determine the optimal IGRT protocol for the individual patient in order to ensure sufficient dose coverage during treatment while reducing imaging dose and frequency remains open.

Due to the persistent lack of recommendations and guidelines, CBCT-based IGRT practice varies profoundly between different institutions. Following a thorough literature research, it can be concluded that most institutions prescind from publishing their CBCT-based IGRT protocols and that the available data originate almost entirely from individual study protocols [7,8,9,10] and a few dedicated surveys from the United States, Canada, Australia and New Zealand [11,12,13,14]. According to these surveys, daily image guidance is frequently used for conventionally fractionated breast radiotherapy, especially when highly conformal treatment techniques and SIB irradiation are employed [11,12,13,14]. Yet, since most patients do not seem to benefit from daily IGRT, a more individual approach is necessary [6]. One possibility of a more individual approach is the determination of positioning inaccuracy during the first three treatment fractions with subsequent adjustment of CBCT frequency (e.g., daily vs. weekly). Even though this approach is already standard procedure in many treatment centers [12,14,15], it lacks reliable evidence. 

The idea of an individual approach based on geometric shifts at the beginning of the treatment stands to reason, as it enables an easy standard operating procedure complying with the recommendations of a case-by-case assessment for IGRT frequency [6,16,17]. Yet, this approach relies on the assumption that the magnitude of geometric shifts during the first three radiotherapy fractions is a good indicator for setup accuracy over the course of treatment and dose distribution in clinical target volumes (CTV) and organs at risk (OAR), which to the best of our knowledge, has not yet been investigated. The aim of the current analysis is to use CBCT data from our patient population treated with adjuvant breast radiotherapy and assess as to what extent the geometric shifts during the first three radiotherapy fractions are able to predict setup accuracy and dose distribution over the course of treatment.

## 2. Materials and Methods

Population and radiotherapy: A total of 45 breast cancer patients receiving adjuvant radiotherapy with daily CBCT imaging (≥27 CBCTs in 28 fractions [Fx]) were retrospectively included in our analysis. All patients were treated in our institute between May 2016 and September 2020. Informed consent was obtained from all participating patients, and the study approval was granted by the local ethics committee. All treatments included whole breast irradiation following lumpectomy and a SIB to the tumor bed. A total of 76% of patients were additionally treated with RNI to the supra- and infraclavicular regions, and 56% of patients were additionally treated with RNI to the internal mammary region. The prescribed dose was 50.4 Gy in 28 Fx to the whole breast and either 58.8 Gy (*n* = 18) or 63 Gy (*n* = 27) in 28 Fx to the tumor bed. The radiotherapy plans were generated in Varian Eclipse 15.6 using VMAT technique according to current guidelines [18,19,20], and the completed plans were approved by a board attending radiation oncologists. For all treatment plans, CTV to PTV margins of 10 mm with exclusion of lung tissue were added, as well as 5 mm for the tumor bed and the lymph node areas.

Positioning and IGRT: Skin markers defined during CT simulation were used for patient setup. The marker-based position was then verified and corrected through daily kilovoltage (kV) CBCT imaging using the Varian On Board Imager 1.6 (Varian Medical Systems, Palo Alto, CA, USA). For each treatment fraction, the CBCT scans were automatically registered to the planning CT using three degrees of freedom and a gray value algorithm, followed by a manual adjustment if necessary. The primary points of interest for manual adjustments were the breast- and boost-CTV as well as the OARs. After three treatment fractions, the position of the skin markers was adjusted using the mean CBCT-based couch shifts along the X-, Y- and Z-axis in order to prevent systematic errors.

Assessment of setup errors: Setup errors were defined as deviation of the marker-based position from the CBCT-adjusted position. By correlating the mean and maximum setup errors in X-, Y- and Z-direction of the first three fractions with the mean deviations of subsequent fractions, we aimed to assess the ability of an early evaluation of position accuracy to predict setup accuracy over the entire course of treatment.

Effect of setup errors on dose distribution: In a second step, we aimed to assess whether mean and maximum setup errors in X-, Y- and Z-direction on treatment days 1–3 correlate with dose coverage in the CTVs and dose to the OARs. For this, a previously established dose accumulation workflow [6] based on deformable image registration (DIR) was adapted for the current study (Figure 1). In a first step, the radiotherapy plans were applied to the respective CBCT scans in Varian Eclipse 15.6 (Varian Medical Systems, Palo Alto, CA, USA) by means of a CBCT-site specific calibration curve. In a next step, deformable image registration (DIR) was utilized to determine deformation vector fields (DVFs) projecting each CBCT scan onto its corresponding planning CT. The resulting dose cubes and DVFs were then exported to an in-house developed script (MATLAB 2019b, The MathWorks Inc., Natick, MA, USA), designed to deform and accumulate the fraction dose cubes. Using this methodology, dose distribution in the CTVs and OARs were simulated for two different scenarios: CBCT-corrected patient setup for all 28 fractions (“gold standard protocol”);CBCT-corrected patient setup for the first 3 fractions and optimized marker-based patient setup for all subsequent fractions (“reference protocol”).

The accumulated dose cubes for both protocols were reimported into Varian Eclipse and compared for dose distribution in the CTVs (breast and tumor bed) and OARs (heart, left anterior descending artery (LAD), ipsilateral lung and contralateral breast). Deviations of dose parameters between the gold standard and reference protocol were assessed and correlated to the setup errors on treatment days 1–3.

Statistical analysis: All statistical analyses were performed using the SPSS statistical analysis software package (SPSS Inc, Chicago, Illinois). Pearson product-moment correlation coefficients were used to assess the correlation between setup errors on treatment days 1–3 and setup errors on subsequent treatment days as well as dose distribution in the CTVs and OARs. Interpretation of correlation coefficients was based on the widely recognized Cohen standard [21].

## 3. Results

### 3.1. Setup Errors

The mean setup errors [range] during the first three radiotherapy fractions (days 1–3) were 0.2 cm [0–0.9 cm] in lateral, 0.2 cm [0–1.5 cm] in cranio-caudal (CC) and 0.2 cm [0–1.3 cm] in anterior–posterior (AP) dimension. The mean setup errors [range] during subsequent fractions (days 4–28) were 0.3 cm [0–1.9 cm] in lateral, 0.4 cm [0–2.1 cm] in CC and 0.3 cm [0–3.2 cm] in AP dimension. No significant correlation was found between mean setup errors on days 1–3 and mean setup errors on days 4–28. Similarly, no significant correlation was found between maximum setup errors on days 1–3 and mean setup errors on days 4–28 in lateral, CC and any dimension, whilst a moderate correlation was found between maximum setup errors on days 1–3 and mean setup errors on days 4–28 in AP dimension. As shown in Figure 2, the magnitude of setup errors increased over the course of treatment. The relation between setup errors on days 1–3 and setup errors on days 4–28 is illustrated in Figure 3.

### 3.2. Correlation of Setup Errors on Days 1–3 and CTV Dose Coverage

For breast-CTV, no significant correlation was found between dose coverage (V95%) and mean setup errors in lateral, AP and any dimension. A moderate correlation was found between dose coverage and mean setup errors in CC dimension. Similarly, no significant correlation was found between dose coverage and maximum setup errors in lateral and AP dimension, whilst a moderate correlation was found between dose coverage and maximum setup errors in CC and any dimension. For SIB-CTV, no significant correlation was found between dose coverage and mean setup errors. Accordingly, no significant correlation was found between dose coverage and maximum setup errors. The relation between CTV coverage and setup errors on days 1–3 is illustrated in Figure 4. Pearson product moment correlation coefficients and the respective *p*-values are summarized in Table 1.

### 3.3. Correlation of Setup Errors on Days 1–3 and OAR Dose Exposure

For heart and LAD, no significant correlation was found between dose exposure (Dmean) and mean setup errors in lateral and CC dimension. A moderate correlation was found between dose exposure and mean setup errors in AP and any dimension. Accordingly, no significant correlation was found between dose exposure and maximum setup errors in lateral and CC dimension, whilst a moderate correlation was found between dose exposure and maximum setup errors in AP and any dimension. For ipsilateral lung and contralateral breast, no significant correlation was found between dose exposure and setup errors. The relation between OAR exposure and setup errors on days 1–3 is illustrated in Figure 5. Product moment correlation coefficients and the respective *p*-values are summarized in Table 2.

## 4. Discussion

Our data indicates that neither mean nor maximum setup errors on treatment days 1–3 can sufficiently predict the magnitude of setup errors during subsequent treatment fractions. In addition, setup errors on treatment days 1–3 correlate poorly with dose coverage of the CTVs and dose exposure to the OARs.

Despite being an increasingly relevant topic in modern adjuvant breast radiotherapy, guidelines on imaging frequency remain sparse. The United States National Comprehensive Cancer Network (NCCN) recommends weekly imaging, yet complementing that under “certain circumstances” more frequent imaging might be appropriate [20]. Not least due to the superficial wording of most guidelines, daily imaging schedules are still commonly used in various studies as well as in daily routine in many treatment centers, especially when highly conformal irradiation techniques such as VMAT and SIB irradiation are employed [11,12,13,14,22]. In our previous study [6], we showed that although daily CBCT remains the highest standard for breast IGRT with SIB irradiation in VMAT technique, a reduction in CBCT frequency from daily to weekly imaging results in only small absolute dose deviations in the CTVs and OARs. In order to reduce imaging dose as well as workload and cost resulting from daily CBCT imaging, we concluded that a more customized approach on breast IGRT is necessary. The crucial question in this context is whether we can identify individual patients at the beginning of treatment who might profit from a more frequent IGRT schedule.

Previous studies have tried to identify risk factors for extensive setup errors, yet none of them managed to establish efficient suggestions on customized breast IGRT. Yang et al. [23] conducted one of the largest studies on this subject on 176 breast cancer patients and found that patients with extensive setup errors on the initial treatment fraction had a high probability of extensive set-up errors during subsequent treatment fractions. As a major limitation, the information on setup errors within their study was based entirely on weekly electronic portal images. In a study by Bertelsen et al. [24] conducted on 77 patients with pulmonary, esophagus or head and neck tumors, CBCT-based information on the patient position during the first three radiotherapy fractions was found not to be representative for the overall patient position. Yet, in this study, CBCT imaging was only performed during the first three treatment fractions and on two further occasions down the treatment sequence. In addition, most previous studies failed to consider the effect of setup errors on CTV coverage and OAR exposure. In our current study, we analyzed data from 45 breast cancer patients having received daily pre-treatment CBCT scans, providing accurate information on setup errors over the course of treatment. The daily CBCT setup allowed for full retrospective 3D dose estimations, enabling us to investigate the effect of setup errors on CTV coverage and OAR exposure. The dose accumulation workflow implemented in our study enables the estimation of dose distribution over the course of treatment in dependence of CBCT frequency. Previous studies consistently acknowledged CBCT-based dose recalculation by means of CBCT site-specific calibration to provide highly accurate dose estimations [23,25]. Despite being associated with a number of uncertainties that might impact our data, DIR and dose accumulation were intensely investigated within previous studies [26,27] and found suitable to be used in commercially available and medically approved software (e.g., MIM Software, Inc., Cleveland, Ohio). All aspects considered, the dose accumulation workflow implemented in our study can therefore safely be considered as the gold standard for retrospective 3D dose estimation. 

Based on our data, it can be concluded that the magnitude of setup errors during the first three treatment fractions is, by itself, not suitable to determine CBCT frequency for the entire treatment. Even large deviations during the first three fractions are not necessarily associated with larger deviations down the treatment course or significant dose deviations in the CTVs and OARs. The CTV-PTV margin has an important impact on CTV coverage. In the current study, a CTV-PTV margin of 10 mm was implemented for the breast, and given the poor correlation between setup errors (Figure 3), it can be assumed that the use of smaller margins would have led to similar results. However, the exact impact of varying CTV-PTV margins on the optimal use of CBCT imaging is beyond the scope of the current trial. Yet, as can be seen in Figure 4, there is still a relevant number of patients that clearly benefit from daily CBCT imaging. Since the analysis of the first three treatment fractions is not a suitable method to detect these individuals, alternative approaches must be considered. One possibility is the daily use of a body-surface scanner, which does not cause additional radiation exposure. So far, however, it remains uncertain whether this approach would provide sufficient positioning accuracy compared to daily CBCT imaging in case of complex target volumes and SIB irradiation. Alternatively, IGRT protocols using CBCT regularly over the course of treatment (e.g., every 3–5 fractions) could be employed. Additionally, the additional use of 2D-based imaging techniques might be an option. Further studies are required to assess the optimum IGRT strategy.

## 5. Conclusions

Neither setup errors over the course of treatment nor dose deviations in the CTVs and OARs correlate sufficiently with the setup errors (mean or maximum deviation in X-, Y- and Z-direction) during the first three treatment fractions. CBCT frequency in adjuvant breast cancer radiotherapy should therefore not be determined solely based on the magnitude of setup errors during the first three treatment fractions.

## Figures and Tables

**Figure 1 cancers-14-04164-f001:**
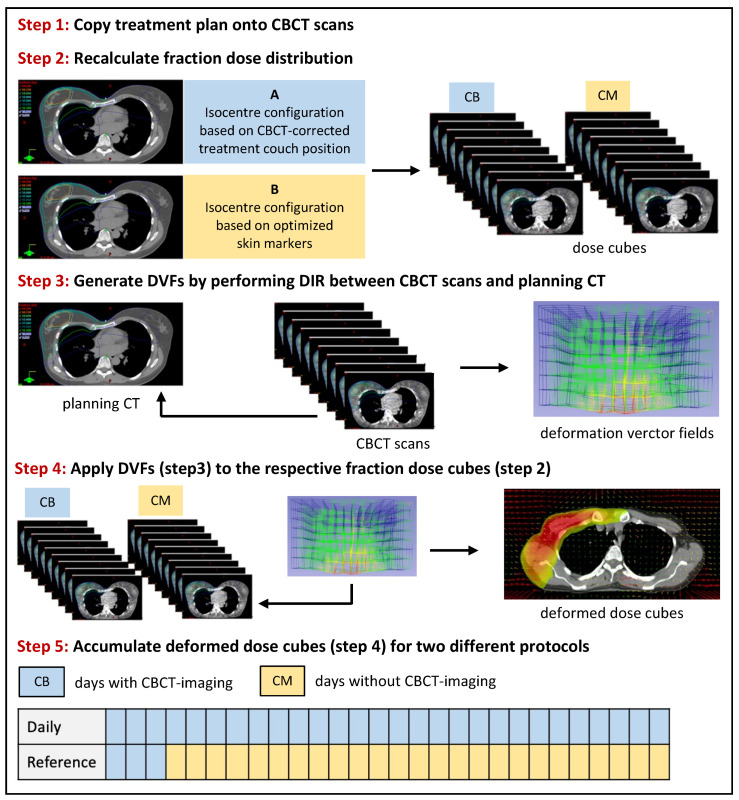
Dose accumulation workflow.

**Figure 2 cancers-14-04164-f002:**
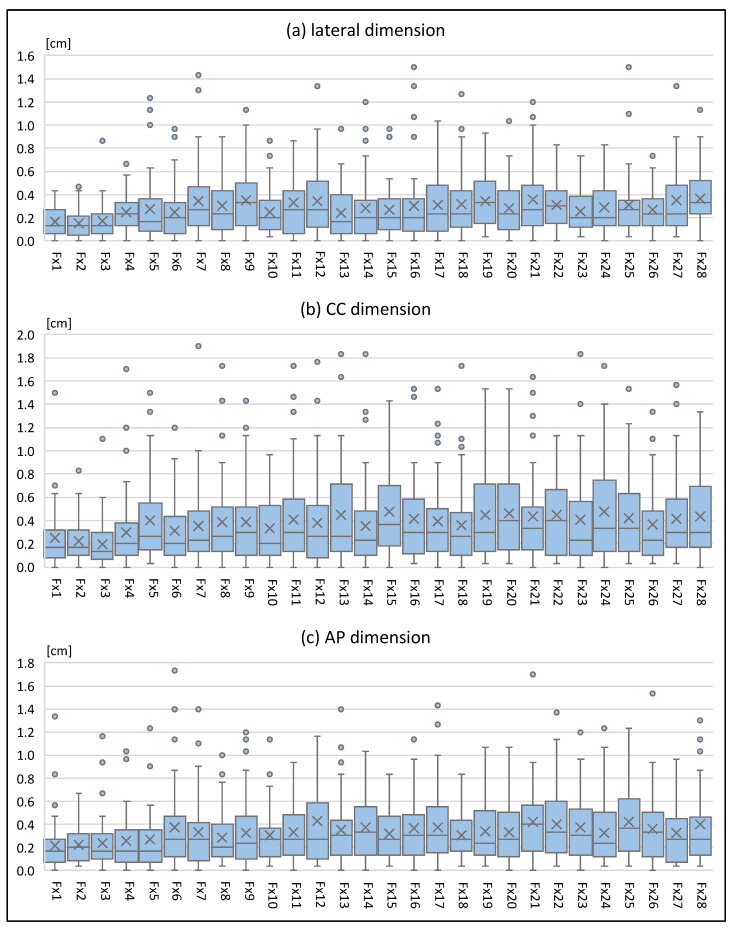
Setup errors in lateral (**a**), CC (**b**) and AP (**c**) dimension during treatment.

**Figure 3 cancers-14-04164-f003:**
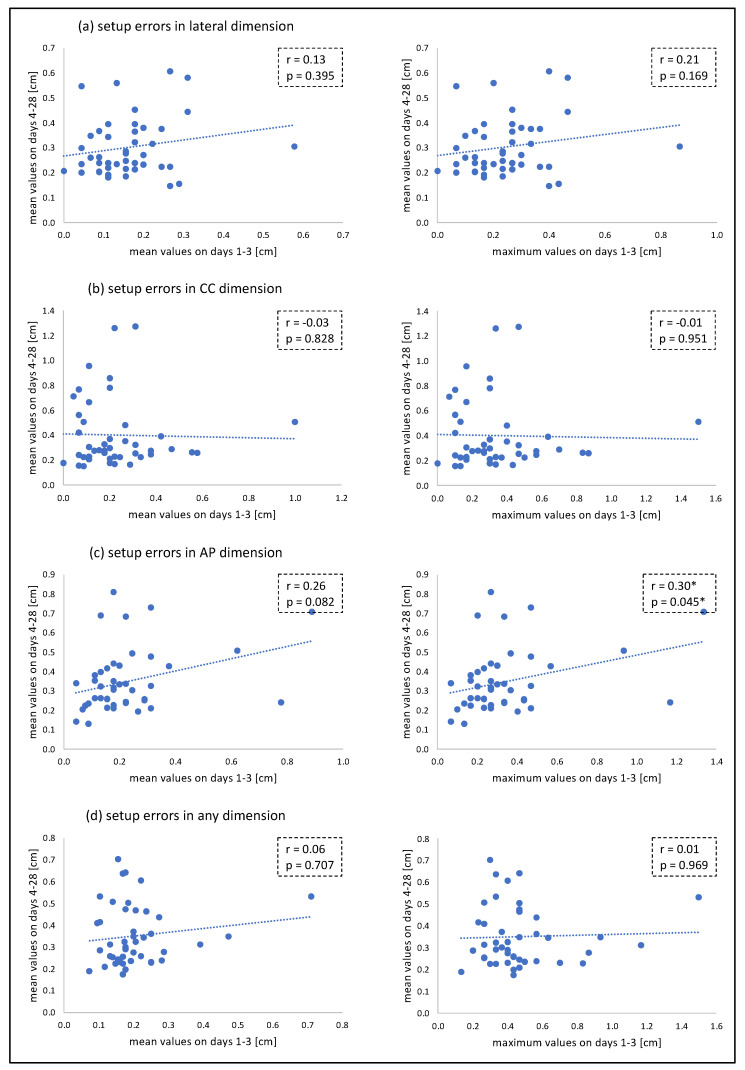
Relation between (mean and maximum) setup errors on days 1–3 and mean setup errors on days 4–28 in lateral (**a**), CC (**b**), AP (**c**) and any (**d**) dimension. * statistically significant (*p* < 0.05).

**Figure 4 cancers-14-04164-f004:**
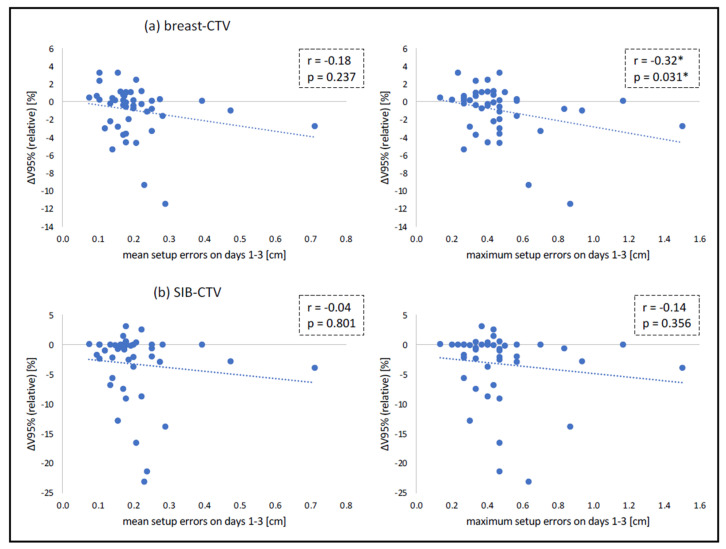
Relation between breast-CTV (**a**) and SIB-CTV (**b**) coverage and (mean and maximum) setup errors in any dimension. * statistically significant (*p* < 0.05).

**Figure 5 cancers-14-04164-f005:**
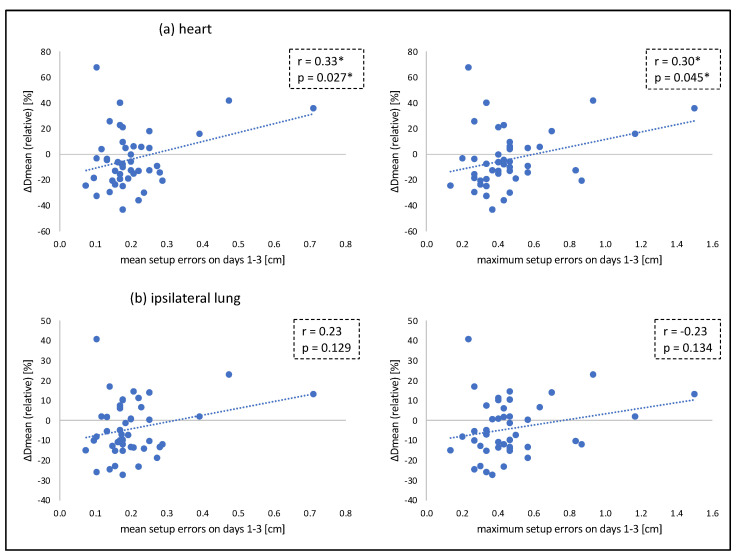
Relation between heart (**a**) and ipsilateral lung (**b**) exposure and (mean and maximum) setup errors in any dimension. * statistically significant (*p* < 0.05).

**Table 1 cancers-14-04164-t001:** Correlation analysis for CTV coverage and setup errors on days 1–3.

	Setup Errors on Days 1–3							
	Any Dimension		Lateral Dimension		CC Dimension		AP Dimension	
V95%	Mean	Max	Mean	Max	Mean	Max	Mean	Max
Breast-CTV	*r* = −0.18	*r* = −0.32 *	*r* = −0.04	*r* = −0.01	*r* = −0.35 *	*r* = −0.37 *	*r* = −0.01	*r* = −0.02
	*p* = 0.237	*p* = 0.031 *	*p* = 0.815	*p* = 0.961	*p* = 0.018 *	*p* = 0.013 *	*p* = 0.993	*p* = 0.879
SIB- CTV	*r* = −0.04	*r* = −0.14	*r* = 0.02	*r* = 0.05	*r* = −0.20	*r* = −0.24	*r* = 0.03	*r* = 0.01
	*p* = 0.801	*p* = 0.356	*p* = 0.886	*p* = 0.720	*p =* 0.196	*p =* 0.106	*p* = 0.860	*p* = 0.989

* statistically significant (*p* < 0.05).

**Table 2 cancers-14-04164-t002:** Correlation analysis for OAR exposure and setup errors on days 1–3.

	Setup Errors on Days 1–3							
	Any Dimension		Lateral Dimension		CC Dimension		AP Dimension	
Dmean	Mean	Max	Mean	Max	Mean	Max	Mean	Max
Heart	*r* = 0.33 *	*r* = 0.30 *	*r* = 0.18	*r* = 0.21	*r* = 0.17	*r* = 0.15	*r* = 0.35 *	*r* = 0.33 *
	*p* = 0.027 *	*p* = 0.045 *	*p* = 0.247	*p* = 0.167	*p* = 0.269	*p* = 0.341	*p* = 0.018 *	*p* = 0.029 *
LAD	*r* = 0.22	r = 0.22	*r* = 0.07	*r* = 0.11	*r* = 0.06	*r* = 0.06	*r* = 0.32 *	*r* = 0.30 *
	*p* = 0.145	*p* = 0.140	*p* = 0.658	*p* = 0.477	*p* = 0.706	*p* = 0.720	*p* = 0.030 *	*p* = 0.044 *
Ips. Lung	*r* = 0.23	r = 0.23	*r* = 0.19	*r* = 0.18	*r* = 0.13	*r* = 0.12	*r* = 0.25	*r* = 0.23
	*p* = 0.129	*p* = 0.134	*p* = 0.221	*p* = 0.226	*p* = 0.413	*p* = 0.442	*p* = 0.098	*p* = 0.126
Con. Breast	*r* = −0.17	r = −0.16	*r* = −0.22	*r* = −0.19	*r* = −0.13	*r* = −0.18	*r* = −0.11	*r* = −0.11
	*p* = 0.256	*p* = 0.293	*p* = 0.142	*p* = 0.222	*p* = 0.379	*p* = 0.249	*p* = 0.473	*p* = 0.484

* statistically significant (*p* < 0.05).

## Data Availability

The data sets generated during and/or analyzed during the current study are available from the corresponding author on reasonable request.

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
