# Peer review of "As Easy as 1, 2, 3? How to Determine CBCT Frequency in Adjuvant Breast Radiotherapy"

_cancers, 2022, doi:10.3390/cancers14174164_

Round 1
Reviewer 1 Report
Interesting paper confirming that accuracy in patient’s positioning is important in adjuvant Breast radiation therapy
Reviewer 2 Report
Thank you for allowing me to review this article.
The authors report data of interest and importance to radiotherapists. However, I cannot entirely agree with the conclusions/interpretation of the results. I also recommend some further clarifications.
Materials and Methods:
I am surprised that the CTV has been expanded by 10mm to the PTV. This, of course, has clear consequences for evaluating the setup errors in terms of "CTV coverage". Why 10mm? Are there any literature references for this? Usually, 5 mm is specified for expansion to compensate for intra- and interfraction positioning errors (see literature: Michalski A et al. Inter- and intra-fraction motion during radiation therapy to the whole breast in supine position: A systematic review. J Med Imag Radiat Oncol 2012;54:499-509).
Results, line 137 ff.
What values are given in the brackets? The range (min-max) or the standard deviations? Please define clearly.
3.2. Correlation of setup error on days 1-3…
You report a "moderate correlation". What is that, and how is it defined, especially in distinction to "significant correlation"? Please define and specify precisely.
4. Discussion
The discussion details that the setup errors during the first 3 fractions do not correlate with the setup errors during the other irradiations. This is, of course, an interesting finding.
However, completely missing from the discussion is that the setup errors were neither correlated with CTV coverage nor OAR exposure. This calls into question the significance of the first finding: -> If the CTV is covered sufficiently anyway and at the same time the OARs are not additionally exposed to radiation (with set up on the first 3 fractions vs. daily CBCT), then – to my mind - it is clinically irrelevant to perform daily CBCT.
In this context, the expansion of the CTV to the PTV must also be discussed, which should not only compensate for the intrafractional motions but also the daily positioning errors. This concept seems to work; as in this study, the positioning errors were not associated with the CTV coverage.
Please discuss these issues in detail.
Round 2
Reviewer 2 Report
All the comments I have made have been adequately answered by the authors and - where necessary - incorporated into the manuscript. Therefore, I believe the manuscript can be published in the present version.